# Assessing international alcohol consumption patterns during isolation from the COVID-19 pandemic using an online survey: highlighting negative emotionality mechanisms

Samantha N Sallie ,[1] Valentin Ritou,[2] Henrietta Bowden-Jones,[1,3] Valerie Voon[1]

► Prepublication history and additional materials for this paper is available online. To view these files, please visit the journal online (http://dx.doi.org/10.1136/bmjopen-2020-044276).

[1]Department of Psychiatry, University of Cambridge, Cambridge, UK
[2]Faculty of Basic and Biomedical Sciences, University of Paris, Paris, France
[3]Faculty of Brain Sciences, University College London, London, UK

**Correspondence to**
Samantha N Sallie;
sns36@cam.ac.uk

## ABSTRACT

**Objectives** The COVID-19 pandemic has required drastic safety measures to control virus spread, including an extended self-isolation period. Stressful situations increase alcohol craving and consumption in alcohol use disorder (AUD) and non-AUD drinkers. Thus, we assessed how COVID-19 related stress may have affected drinking behaviours in the general population.

**Design** We developed an online cross-sectional survey, Habit Tracker (HabiT), which measured changes in drinking behaviours before and during COVID-19 quarantine. We also assessed psychiatric factors such as anxiety, depression (Hospital Anxiety and Depression Scale) and impulsivity (Short-Impulsive Behavior Scale). Lastly, we related drinking behaviours to COVID-19 specific stress factors.

**Setting** HabiT was released internationally, with individuals from 83 countries participating.

**Participants** Participants were included if they were 18 years of age or older and confirmed they were proficient in English. The survey was completed by 2873 adults with 1346 usable data (46.9% accurately completed).

**Primary outcome measures** Primary outcome measures were change in amount and severity of drinking behaviours before and during quarantine, and current drinking severity during quarantine.

**Results** Although drinking behaviours decreased overall during quarantine, 36% reported an increase in alcohol use. Those who increased alcohol use during quarantine were older individuals (95% CI 0.04 to 0.1, p<0.0001), essential workers (95% CI −0.58 to −0.1, p=0.01), individuals with children (95% CI −12.46 to 0.0, p=0.003), those with a personal relationship with someone severely ill from COVID-19 (95% CI −2 to −0.38, p=0.01) and those with higher depression (95% CI 0.67 to 1.45, p<0.0001), anxiety (95% CI 0.61 to 1.5, p=0.0002), and positive urgency impulsivity (95% CI 0.16 to 0.72, p=0.009). Furthermore, country-level subsample analyses indicated that drinking amount (95% CI 9.36 to 13.13, p=0.003) increased in the UK during quarantine.

**Conclusions** Our findings highlight a role for identifying those vulnerable for alcohol misuse during periods of self-isolation and underscore the theoretical mechanism of negative emotionality underlying drinking behaviours driven by stress. Limitations include a large degree of

## Strengths and limitations of this study

► The Habit Tracker (HabiT) study sampled drinking behaviours of a large, diverse population during the COVID-19 pandemic.
► Changes in drinking behaviours were assessed against specific COVID-19 related stress factors.
► Due to the length of the survey (8–10 min), we observed a large degree of study dropout.
► Subjects were within varying phases of lockdown during the time of testing.
► The prevalence of diagnosed alcohol use disorder drinkers sampled was low, likely related to sampling issues or under-reporting.

study dropout (n=1515). Future studies should assess the long-term effects of isolation on drinking behaviours.

## INTRODUCTION

The COVID-19 pandemic has necessitated drastic safety measures to control the virus spread. These measures included an extended self-isolation period in which individuals were permitted to leave their places of residence only to obtain amenities (eg, food, medical care, and toiletries) or engage in essential work. Individuals were not permitted face-to-face contact with anyone who did not reside within their immediate households. In the UK, these measures were instituted nationally on 23 March 2020, with a gradual lifting of restrictions on 10 May 2020, ending on 4 July 2020 with locality-specific intermittent reinstatement of these measures. Although a necessary precautionary measure to mitigate the devastating effects of COVID-19 on public health, evidence indicates that protracted periods of self-isolation, especially in the context of stress, may be related to acute and prolonged negative mental health

consequences, particularly in individuals struggling with psychiatric disorders.[1]

Indeed, current clinical reports from individuals in treatment for substance abuse disorder indicate that the stress produced by COVID-19 social isolation measures have triggered greater and more frequent drug or alcohol cravings, subsequently leading to relapse.[2] This observation is relevant to a prominent mechanistic theory of negative emotionality underlying alcohol misuse.[3] The relationship between stress and alcohol consumption is widely recognised and can be observed in an experimental fashion.[4] In subjects with known alcohol use disorder (AUD), stress and experimental manipulations of stress enhance the amount of alcohol consumed,[5 6] alcohol craving,[7] problematic drinking behaviours, and likelihood of relapse.[8] Exposure to stress triggers relapse characterised by a reinstantiation of alcohol cravings and alcohol-seeking behaviours.

Increases in alcohol craving and consumption after stress exposure also occur in those without AUD. An increase in alcohol consumption is often used as a coping strategy for both chronic and specific stressful life events in both AUD and non-AUD drinkers.[9] Similarly in both groups, self-reported craving and subjective judgements of alcohol value rise following a stress task,[10] and social drinkers consume more alcohol after witnessing a social stressor.[11] These relationships are moderated by gender,[12] age,[13] previous alcohol exposure,[13] alcohol expectancies,[14] and the pattern of alcohol consumption.[15] Furthermore, psychiatric symptomology such as anxiety and depression as well as pathological levels of personality traits such as impulsivity are widely recognised predisposing factors to problematic alcohol use and addiction.[3 16]

Thus, in response to these exceptional circumstances, we aimed to assess how social isolation measures in the midst of the COVID-19 pandemic may have affected drinking behaviours in the general adult population. We developed an international survey, entitled Habit Tracker (HabiT), which evaluated drinking severity before (post-hoc recall) and during the COVID-19 quarantine period. We hypothesised that changes in amount of alcohol consumption and severity of drinking behaviours may be related to specific COVID-19 related stress factors, as well as demographic and psychiatric factors. Furthermore, we investigated if COVID-19 related stress factors influenced changes in drinking amount, drinking severity, depression, and anxiety before and during quarantine.

## METHODS
### Recruitment and inclusion criteria
The HabiT survey was a questionnaire that sought to assess the effects of isolation on alcohol, smoking, and internet use. The effects on alcohol use are reported here. Subjects were included if they were 18 years of age or older and confirmed they were proficient in reading and understanding English. HabiT was advertised by University of Cambridge news page on 11 May 2020, a day before its international release. For the next several days, the survey was disseminated by news agencies throughout the UK (eg, The Telegraph, BBC Cambridgeshire and News Wise) as well as throughout various University of Cambridge colleges. Furthermore, the survey was posted and shared on personal and public social media sites (ie, Facebook and Twitter). All subjects gave informed consent and were not financially compensated for their participation, although informed that—on survey completion–they would be provided results of the study through request. The data collected was fully anonymised. The survey was created using Qualtrics (Provo, Utah) survey-building platform. Developed iteratively within-lab and among coauthors to insure brevity and consistency, the average time to complete the survey was approximately 8–10 min, and all subjects could participate on either a computer or smartphone device.

### Patient and public involvement statement
We did not involve patients or the public in the research design, reporting, or survey dissemination strategies of this study.

### Demographic information
The demographic information collected were as follows: age, gender, socioeconomic status, intimate relationship status, country and city of residence, and any previous or current diagnosis of a psychiatric or neurological disorder.

### Attentional checks
Every major section of the survey contained at least one question that served as an attentional check to ensure subjects were correctly reading and answering survey questions to the best of their ability. The attentional checks were structured to mirror the Likert scaling of each section (eg, 'If you are reading this question, please select "Strongly Agree."').

### Frequency and severity of alcohol consumption before and during the quarantine period
We first asked subjects if they drank alcohol. If the answer was negative, they proceeded to the next set of questions. If the answer was positive, we assessed the change in the amount and severity of alcohol use as well as the current severity of alcohol use. We asked subjects to report the following behaviours within a typical week in November (ie, pre-quarantine) and within the last week (ie, during quarantine): (1) the number of units of alcohol consumed within the last week with examples for the number of units for differing types of alcohol and sizes provided; and (2) the change in severity using a timescale adaptation of the first three questions of the Alcohol Use Disorders Identification Test, which assessed the amount and frequency of alcohol consumption (AUDIT-C).[17] Subjects were asked to report how many days in the last week they consumed an alcoholic beverage, how many drinks they consumed on a typical day they were drinking in the last week, and how often they consumed

six or more drinks on one occasion in the last week. To assess the current severity of drinking behaviours during quarantine, we used a timescale-adapted version of the full Alcohol Use Disorders Identification Test (AUDIT),[18] which assessed problem drinking behaviours within the last week such as an inability to stop drinking once started, failure to perform responsibilities, feeling guilt or remorse, drinking shortly after waking to ease the adverse physiological effects of drinking the night before, drinking to the point of memory loss, injuring oneself or others due to drinking, and concern from a loved one or medical professional related to the amount or severity of one's drinking. We used two primary outcome measures: the change in severity (AUDIT-C), corroborated with the secondary change in amount of drinking (units per week) and current severity (full AUDIT).

## COVID-19 related stress factors

We assessed 10 factors that may contribute to COVID-19 related stress using the following questions:
1. Have you been deemed an 'essential worker' by your government?
2. Do you work for healthcare services specifically with individuals who have contracted COVID-19? (Subquestion of question 1)
3. Has your employment situation changed due to the COVID-19 crisis?
4. Has anyone you know personally contracted or have shown symptoms characteristic of COVID-19?
5. Has anyone you know personally become severely ill or died due to contracting COVID-19?
6. Are you isolated alone?
7. Do you have children?
8. If you have children, are you their only caretaker? (Subquestion of question 7).
9. If you are currently in isolation with others, how would you describe the quality of your relations?
10. How often do you currently go outdoors (for work, essential duties, leisure and so on)?

## Psychiatric measures

Depression and anxiety symptomatology were measured using The Hospital Anxiety and Depression Scale (HADS), a brief, validated four-item questionnaire.[19] As a secondary analysis, we assessed impulsivity using the validated Short Impulsive-Behavior Scale (SUPPS-P).[20] This scale provides an overall impulsivity score, as well as five scores corresponding to impulsivity subscales: perseveration, lack of premeditation, sensation seeking, negative urgency, and positive urgency.

## Statistical analysis

Statistical analyses were performed using MATLAB (V.2020a). All subjects who answered the attentional checks incorrectly (n=12), reported highly improbable answers regarding the units of alcohol they consumed weekly (eg, 1000 units), did not report their gender, or did not complete the psychiatric questionnaires were excluded from further analysis; leaving a total of 1346 subjects. Drinking severity scores of the sample were non-normally distributed (Shapiro-Wilk, $p<0.05$), thus non-parametric tests were used.

We used Mann-Whitney U tests to compare weekly alcohol unit consumption and alcohol severity before and during quarantine in the full group. Then, we divided subjects into three groups, those who during quarantine either increased, decreased, or did not change their alcohol consumption, and performed a Kruskal-Wallis H-test to assess the relative drinking amount to severity indices of these groups.

We then assessed which COVID-19 related stress factors were associated with changes in either amount (alcohol units consumed per week), change in severity (AUDIT-C), current severity (full AUDIT), or current depression and anxiety (HADS) using the following tests: (1) Mann-Whitney U tests to compare negative versus positive responses to the COVID-19 stress factors (MW), (2) multivariate analysis of covariance (MANCOVA)[21] controlling for gender and age (MAN1), and (3) a second MANCOVA controlling for age, gender, depression, and anxiety symptomology (MAN2). For the MANCOVA tests, variables 'age,' 'depression severity,' and 'anxiety severity' were dichotomised via median split. For the COVID-19 stress primary factor comparisons (eight items), we used False Discovery Rate (FDR) to control for multiple comparisons with significance assigned at $p<0.05$.[22 23] 95% confidence intervals (CIs) are provided with p-values for significant findings observed from the most stringent statistical test.

On an exploratory basis, we then used Spearman's partial correlations to compare the drinking severity indices of subjects who completed the timescale-adapted full AUDIT with SUPPS-P and HADS scores to relate drinking severity of the overall sample to psychiatric measures. Lastly, in order to assess possible directional relationships in changes in the severity of drinking behaviours to depression, anxiety, and impulsivity, we performed Spearman's partial correlations with the psychiatric questionnaires among the three aforementioned groups (ie, increased, decreased, and null). For both correlational analyses, we used FDR correction ($p<0.05$) for multiple comparisons.

## RESULTS

### Demographic information

A total of 2873 subjects participated (data collection: 12 May 2020–28 May 2020) of which 1346 had usable data based on defined criteria (1515 dropouts; 46.9% accurately completed; please refer to the supplementary materials for a demographic analysis of those who did not complete the survey). Of these subjects, 859 (63.8%) reported that they drink alcohol (please refer to the online supplemental materials for demographic information on those who reported drinking alcohol). Of the 1346 subjects, the average age was 28.92±10.45 years (95% CI 28.2 to 29.53) (range=18–90) with more males (males: n=1006; females: n=325; other: n=15) from 85 different

countries of residence, with the majority from the UK (n=434) and the USA (n=355), followed by Canada (n=64) and Germany (n=63). Marital status was as follows: single: n=785; married or committed: n=571; divorced or separated: n=33; widowed: n=4. Socioeconomic status (as denoted by annual income in raw currency on the country level and converted to UK pounds during analysis) was as follows: <19.9k: n=285; 20–39.9k: n=273; 20–39.9k: n=244; 40–69.9k: n=241; 70–99.9k: n=141;>100k: n=203; and 232 subjects did not report their incomes. Current psychiatric or neurological diagnoses were as follows: no diagnosis: n=1192; depression: n=60; anxiety: n=38; post-traumatic stress disorder: n=5; and comorbid depression and anxiety: n=46.

## Overall changes in drinking frequency and severity before and during quarantine

Of the total sample, the change in problem drinking severity (AUDIT-C) was 0.89±1.43 (95% CI 0.81 to 0.96) (range: 0–8) and the mean change in the amount consumed was 5.62±9.55 units per week (95% CI 3.16 to 4.02) (range: 0–120). The current problem drinking severity (full AUDIT) was 3.14±4.47 (95% CI 2.9 to 3.37) (range: 0–32), with 557 subjects included that do not consume alcohol. Of the subjects who reported they consume alcohol (n=859),

the change in severity from prequarantine to quarantine was a decrease of 1.53±1.6 (95% CI 5.01 to 5.64), range 0–8 (U=2.65 (95% CI 0 to 0.21) p=0.008). The units of alcohol consumed per week was significantly decreased during the quarantine period (8.03±14.22 units (7.11–8.94) range=1–120) compared with November (8.32±11.92 units (95% CI 7.47 to 9.02) range=0–150), U=−2.29 (95% CI 0.0 to 0.0) p=0.02 (figure 1). However, in the UK, the units of alcohol consumed per week was significantly increased during the quarantine period (11.25±17.73 units (95% CI 9.36 to 13.13) range=1–120) compared to November (10.94±14.17 units (95% CI 9.44 to 12.45) range=0–150), U=3.0 (95% CI 0 to 0.7) p=0.003. (For full country-level subanalyses of drinking behaviours, as well as severity of lockdown and amount of confirmed COVID-19 cases and deaths during the data collection period by country via Coronavirus Government Response Tracker,[24] please refer to the online supplemental materials). Of the international sample, 172 (20%) subjects reported abstention from alcohol consumption during the quarantine period. More subjects reported a decrease (n=384, 45%) or an increase (n=308, 36%) as opposed to no change (n=166, 19%) of weekly alcohol consumption from November to the quarantine period ($\chi^2$=72.86, p=0.001; figure 1). Of the three

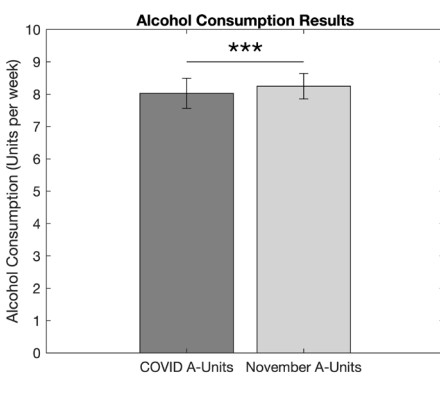
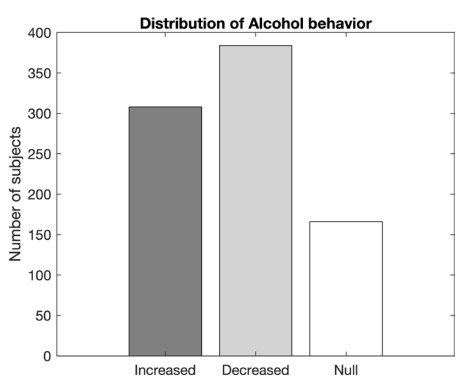
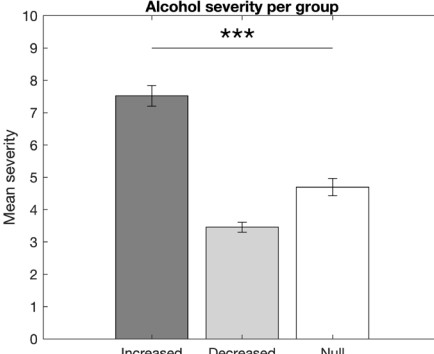
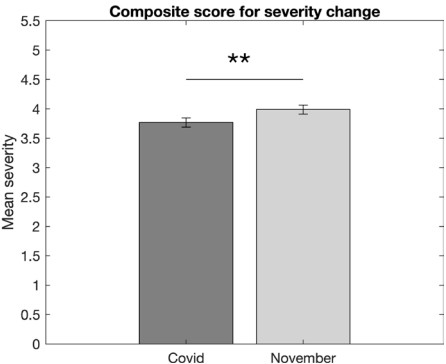

**Figure 1** Changes in amount and severity of drinking behaviours in the HabiT sample between prequarantine and quarantine periods. Units of alcohol consumed weekly (top left) and changes in drinking severity (AUDIT-C) (bottom right) decreased during the quarantine period and more individuals either increased or decreased their weekly units consumed during quarantine than remained the same (top right). Furthermore, those who increased their weekly alcohol unit consumption during the quarantine period had significantly higher drinking severity indices (full AUDIT) compared with those who decreased or did not change their drinking behaviours during the quarantine period (bottom left). AUDIT, Alcohol Use Disorders Identification Test; HabiT, Habit Tracker.

groups, those who: (1) increased weekly units consumed during quarantine (7.5±10.5 change in units (95% CI 6.33 to 8.7) range: 1–80), (2) decreased weekly units consumed during quarantine (–6.5±9.5 change in units (95% CI –7.45 to –5.55) range: –0.2 to –120), and (3) did not change their weekly unit consumption, subjects who had increased the units of alcohol consumed during the quarantine period showed significantly higher current drinking severity scores (7.5±5.6 (95% CI 6.89 to 8.15) range: 1–32) than those who reported decreases (3.5±3.0 (95% CI 3.16 to 3.76) range: 1–21) or no changes (4.8±3.6 (95% CI 4.17 to 5.23) range: 1–20) in weekly unit consumption (H=165.33 (95% CI 3.35 to 4.78) p<0.0001, figure 1).

## COVID-19 stress factor evaluation

The change in amount of drinking was positively correlated with age ($r_s$=0.2 (95% CI 0.04 to 0.1) p<0.0001) and gender with males (6.44±10.8 units (95% CI 5.63 to 7.35) range: 0–120) showing an increased change in drinking amount relative to females (3.81±5.18 (95% CI 3.08 to 4.32) range: 0–38), or other genders (1.32±1.65 (95% CI 0.18 to 2.24) range: 0–5) (H=8.17, p=0.003). Changes in drinking severity were also related to both age and gender, with older individuals ($r_s$=0.2 (95% CI 0.01 to 0.02) p<0.0001) and males (1.68±1.74 (95% CI 1.55 to 1.83) range: 0–8) demonstrating greater changes in their drinking severity than females (1.16±1.12 (95% CI 1.02 to 1.3) range: 0–8), and others (1.36±1.29 (95% CI 0.54 to 2.18) range: 0–3) (H=6.02 (95% CI –0.81 to –0.22) p=0.05). (Age- and gender-specific subanalyses of drinking behaviours can be found in the online supplemental materials). Thus, we used age and gender as covariates for both MANCOVA analyses. All relevant covariates used in these analyses were dichotomised via median split (age=25.1 years, depression severity=2, and anxiety severity=1).

### Primary COVID-19 stress factors

The influence of COVID-19 stress factors on the change in drinking severity, amounts consumed, and current drinking severity are reported in tables 1–3, respectively. Designated essential workers and those with children showed a greater increase in the amount consumed weekly, drinking severity, and greater current severity. This remained significant including when controlled for demographic variables (age and gender) and psychiatric symptoms (depression and anxiety). Notably, although subjects with children reported an increase in the number of units of alcohol and severity of alcohol use, they also reported lower levels of depression and anxiety. Knowing an individual personally who was ill or severely ill with COVID-19 showed higher current alcohol drinking severity than those who did not, but with no change from prequarantine to postquarantine. A reported change in employment status and isolating alone was associated with greater depression scores, with no differences in drinking behaviours. Isolating with others but reporting a poor quality relationship was associated with greater depression and anxiety; however, the lower drinking behaviours were moderated by age and gender effects. Finally, going outdoors was associated with greater current drinking severity and greater depression and anxiety scores controlling for all variables. Post hoc tests confirmed that, in cases in which a significant relationship was lost between an item and either changes in drinking amount or severity due to controlling for age and gender (ie, MANCOVA 1), age was the sole contributor (essential worker: F(1, 533.2)=7 (95% CI 0.15 to 2.1) p=0.008; others ill: F(1, 879.9)=52.6 (95% CI 1.7 to 2.7) p<0.0001; poor relationship: F(1, 933.9)=48.88 (95% CI 1.8 to 2.8) p<0.0001).

### Secondary COVID-19 stress factors

Two COVID-19 stress factors were considered secondary as they represented a subset of a primary factor. Working for healthcare services was associated with a trend towards

**Table 1** COVID-19 primary stress items relationship with changes in drinking severity (as indexed by the AUDIT-C) from prequarantine to quarantine.

| Stress factor | N total | Yes M (SD) | N Yes | No M (SD) | N No | MW P value | MAN1 P value | MAN2 P value | 95% CI |
|---|---|---|---|---|---|---|---|---|---|
| Essential worker | 1337 | 0.16 (1.9) | 241 | –0.21 (1.6) | 1096 | 0.02* | 0.01* | 0.01* | –0.58 to –0.1 |
| Employment | 1337 | –0.14 (1.8) | 323 | –0.14 (1.6) | 1014 | 0.83 | 0.96 | 0.92 | |
| Others ill | 1334 | –0.17 (1.8) | 497 | –0.12 (1.6) | 837 | 0.75 | 0.64 | 0.63 | |
| Others severely ill | 1336 | –0.01 (2) | 127 | –0.15 (1.6) | 1209 | 0.35 | 0.7 | 0.69 | |
| Isolated alone | 1325 | –0.1 (1.9) | 168 | –0.15 (1.6) | 1157 | 0.83 | 0.85 | 0.82 | |
| Having children | 1334 | 0.34 (1.4) | 209 | –0.23 (1.7) | 1125 | <0.0001* | 0.005* | 0.003* | –12.46 to 0.0 |
| Poor relationship | 1168 | –0.3 (1.7) | 187 | –0.13 (1.6) | 981 | 0.35 | 0.7 | 0.69 | |
| Going outdoors | 1336 | –0.27 (1.3) | 193 | –0.12 (1.7) | 1143 | 0.26 | 0.7 | 0.69 | |

95% confidence interval (CI) for most stringent statistically significant finding.
*p-value indicates statistical significance.
AUDIT-C, Alcohol Use Disorders Identification Test; M, mean; MAN1 p value, MANCOVA p-value controlling for age and gender; MAN2 p-value, MANCOVA p-value controlling for age, gender, depression, and anxiety; MW p-value, Mann-Whitney U-Test p-value; SD, standard deviation.

**Table 2** COVID-19 primary stress items relationship with changes in drinking amount (in units) from prequarantine to quarantine.

| Stress factor | N total | Yes M (SD) | N Yes | No M (SD) | N No | MW P value | MAN1 P value | MAN2 P value | 95% CI |
|---|---|---|---|---|---|---|---|---|---|
| Essential worker | 1337 | 1.26 (12.8) | 241 | 0.45 (7.5) | 1096 | 0.0003* | 0.07 | 0.08 | −3.4 to −0.02 |
| Employment | 1337 | 0.17 (11.2) | 323 | 0.13 (7.8) | 1014 | 0.77 | 0.95 | 0.97 | |
| Others ill | 1334 | 0.05 (7.1) | 497 | 0.2 (9.6) | 837 | 0.83 | 0.95 | 0.97 | |
| Others severely ill | 1336 | 0.06 (7.6) | 127 | 0.15 (8.9) | 1209 | 0.83 | 0.95 | 0.97 | |
| Isolated alone | 1325 | 0.05 (11.6) | 168 | 0.2 (8.2) | 1157 | 0.46 | 0.95 | 0.97 | |
| Having children | 1334 | 2.02 (11.9) | 209 | 0.54 (7.9) | 1125 | <0.0001* | 0.04* | 0.02* | −3.6 to −0.74 |
| Poor relationship | 1168 | 0.4 (6.1) | 187 | 0.19 (8.7) | 981 | 0.46 | 0.95 | 0.97 | |
| Going outdoors | 1336 | 1.23 (6.8) | 193 | 0.04 (9.0) | 1143 | 0.15 | 0.47 | 0.4 | |

95% confidence interval (CI) for most stringent statistically significant finding.
*p-value indicates statistical significance.
M, mean; MAN1 p-value, MANCOVA p-value controlling for age and gender; MAN2 p-value, MANCOVA p-value controlling for age, gender, depression, and anxiety; MW p-value, Mann-Whitney U-Test p-value; SD, standard deviation.

**Table 3** COVID-19 primary stress items relationship with current drinking severity (ie, full AUDIT), depression and anxiety from prequarantine to quarantine.

| Stress factor | N Total | Severity type | Yes M (SD) | N Yes | N M (SD) | N No | MW P value | MAN1 P value | MAN2 P value | 95% CI |
|---|---|---|---|---|---|---|---|---|---|---|
| Essential worker | 1337 | Drinking | 4.42 (5.7) | 243 | 2.85 (4.1) | 1099 | <0.0001* | 0.0005* | 0.0005* | −1.8 to −057 |
| | | Depression | 2.29 (1.8) | 243 | 2.44 (1.9) | 1099 | 0.43 | 0.84 | | |
| | | Anxiety | 1.79 (1.7) | 243 | 1.94 (1.8) | 1099 | 0.42 | 0.8 | | |
| Employment change | 1337 | Drinking | 3.46 (4.9) | 324 | 3.02 (4.3) | 1018 | 0.38 | 0.08 | 0.144 | |
| | | Depression | 2.78 (2.0) | 324 | 2.31 (1.9) | 1018 | 0.0043* | 0.007* | | −0.58 to −0.1 |
| | | Anxiety | 2.03 (4.5) | 324 | 1.88 (1.8) | 1018 | 0.32 | 0.363 | | |
| Others ill | 1334 | Drinking | 3.59 (1.9) | 499 | 2.87 (4.4) | 840 | <0.0001* | 0.1 | 0.125 | −1.2 to −0.2 |
| | | Depression | 2.3 (1.8) | 499 | 2.47 (1.9) | 840 | 0.20 | 0.83 | | |
| | | Anxiety | 1.9 (5.5) | 499 | 1.93 (1.9) | 840 | 0.99 | 0.94 | | |
| Others severely ill | 1336 | Drinking | 4.49 (2.0) | 127 | 2.99 (4.3) | 1214 | 0.001* | 0.007* | 0.01* | −2 to −0.38 |
| | | Depression | 2.45 (2.0) | 127 | 2.4 (1.9) | 1214 | 0.99 | 0.41 | | |
| | | Anxiety | 1.92 (5.8) | 127 | 1.91 (1.8) | 1214 | 0.82 | 0.84 | | |
| Isolated alone | 1325 | Drinking | 3.88 (2.0) | 169 | 2.98 (4.2) | 1161 | 0.42 | 0.83 | 0.87 | |
| | | Depression | 3.4 (1.9) | 169 | 2.41 (1.9) | 1161 | 0.009* | 0.04* | | −0.7 to −0.06 |
| | | Anxiety | 2.04 (5.2) | 169 | 1.9 (1.8) | 1161 | 0.43 | 0.11 | | |
| Having children | 1334 | Drinking | 5.17 (1.8) | 211 | 2.75 (4.2) | 1128 | <0.001* | 0.0003* | <0.0001* | −2.4 to −0.9 |
| | | Depression | 1.5 (1.7) | 211 | 2.58 (1.9) | 1128 | <0.0001* | <0.0001* | | 0.37 to 0.97 |
| | | Anxiety | 1.37 (1.7) | 211 | 2.02 (1.9) | 1128 | <0.0001* | 0.0009* | | 0.25 to 0.85 |
| Poor relationship | 1168 | Drinking | 2.82 (5.1) | 187 | 3.1 (4.1) | 985 | 0.01* | 0.92 | 0.87 | 0.4 to 1.0 |
| | | Depression | 3.57 (2.0) | 187 | 2.2 (1.8) | 985 | <0.0001* | <0.0001* | | −1.53 to −1 |
| | | Anxiety | 2.79 (2.0) | 187 | 1.74 (1.8) | 985 | <0.0001* | <0.0001* | | −1.3 to −073 |
| Going outdoors | 1336 | Drinking | 3.42 (4.5) | 1148 | 1.37 (3.4) | 193 | <0.0001* | <0.0001* | <0.0001* | 1.14 to 2.47 |
| | | Depression | 3.18 (2.0) | 193 | 2.28 (1.9) | 1148 | <0.0001* | <0.0001* | | −1 to −0.42 |
| | | Anxiety | 2.42 (2.0) | 193 | 1.83 (1.8) | 1148 | 0.0002* | 0.0008* | | −0.8 to −0.24 |

95% confidence interval (CI) for most stringent statistically significant finding.
*p-value indicates statistical significance.
AUDIT, Alcohol Use Disorders Identification Test; M, mean; MAN1 p-value, MANCOVA p-value controlling for age and gender; MAN2 p-value, MANCOVA p-value controlling for age, gender, depression, and anxiety; MW p-value, Mann-Whitney U-Test p-value; SD, standard deviation.

a greater change in amount of units consumed (F=3.97 (95% CI −6.73 to −0.0), p=0.05) and greater severity of current drinking (F=7.01 (95% CI −3.9 to −0.6) p=0.007) when controlled for all variables. Being the only caretaker for children was also associated with greater change in drinking severity (U=2.62 (95% CI −2.7 to −0.9) p=0.009) and greater change of amount consumed (U=2.67 (95% CI −4.5 to −0.8) p=0.007) but was no longer significant when controlling for age and gender.

### Drinking severity during quarantine and correlations with psychiatric measures

Of the individuals who reported drinking alcohol, (n=769) completed the current drinking severity index (eg, the adapted-timescale full AUDIT). The severity of drinking behaviours was positively related to depression ($r_s$=0.12 (95% CI 0.34 to 0.79) p=0.004), anxiety ($r_s$=0.12 (95% CI 0.3 to 0.74) p=0.027) and positive urgency impulsivity ($r_s$=0.12 (95% CI 0.14 to 0.34) p=0.004), controlled for age and gender. To assess potential directional relationships between current drinking severity during quarantine and psychiatric measures, we correlated depression, anxiety, and impulsivity with the three drinking groups (ie, increased, decreased and null). Drinking severity scores in the decreased and no change groups were not significantly correlated with any of the psychiatric measures of interest. However, drinking severity of those who increased their units consumed during the quarantine period were related to depression ($r_s$=0.30 (95% CI 0.67 to 1.45) p<0.0001), anxiety ($r_s$=0.23 (95% CI 0.61 to 1.5) p=0.0002), and positive urgency ($r_s$=0.17 (95% CI 0.16 to 0.72) p=0.009) (figure 2).

### DISCUSSION

We show an overall decrease in amounts and severity of problem alcohol use from prequarantine to the quarantine period. Critically, however, three different subpopulations were identified, with most either increasing or decreasing use as compared with remaining unchanged in their alcohol use behaviours. Greater drinking

was associated with demographic factors including age, COVID-19 stress-related factors, and psychiatric factors such as depression, anxiety, and the impulsivity subscale of positive urgency. Increases in drinking were also region-specific; with UK residents demonstrating an upswing in weekly amount of alcohol consumed during quarantine. Our findings underscore the theoretical mechanism of negative emotionality underlying drinking behaviours driven by stress, depression and, anxiety.

An overall decrease in alcohol consumption and problematic use may have multiple, potential aetiologies. Stringent lockdown may be associated with a decrease in the presence or availability of alcoholic beverages within the immediate household given limitations in access, a decrease in exposure to alcohol cues that may trigger urges, or the preference to consume alcohol within social contexts. More subjects reported either decreasing or increasing the frequency of their alcohol intake compared to remaining unchanged, consistent with previous reports of a greater tendency towards extremes in individual drinking patterns when faced with either acute or chronic life stressors.[15]

Older individuals showed a greater increase in drinking behaviours during lockdown and current severity of problem drinking, consistent with demographic factors known to be associated with alcohol misuse. Whether one increases their drinking after experiencing acute or chronic life stress is age dependent, which may reflect a function of previous alcohol experience.[13] Age may play a particularly unique role in the context of COVID-19 due to the greater need for stringent isolation with age, potentially fewer supports, and the risk of greater isolation, loneliness, and concern about the impact of COVID-19 on one's personal health. Expectedly, males showed greater unit consumption compared with females and other genders overall. However, males showed a decrease in both drinking amount and severity during quarantine, while females demonstrated the opposite trend. This finding corroborates evidence that indicates females are

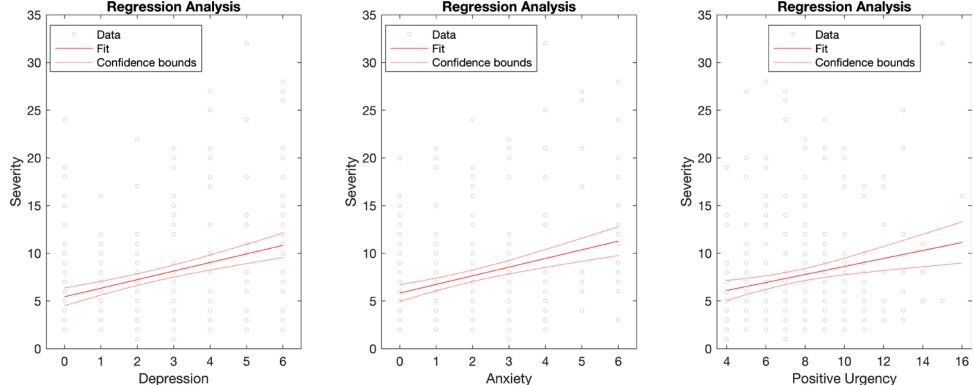

**Figure 2** Regression plots of the significant relationships between drinking severity and psychiatric measures in subjects who increased weekly alcohol unit consumption during quarantine. Drinking severity indices of the group who increased their drinking during the quarantine period were significantly positively related to depression severity, anxiety severity and positive urgency (impulsivity subset).

more likely than males to consume alcohol in order to cope with stress.[25]

COVID-19 specific stress factors appear to influence drinking behaviours controlling for other confounding variables. Being deemed an essential worker and having children was associated with a greater increase in drinking behaviours during quarantine. Importantly, although having children was associated with an increase in alcohol use, depression and anxiety scores were lower than in those without children. This suggests the additional burden of childcare and home schooling contributed to the tendency towards drinking- possibly in the context of stress relief- and was not mediated by greater depression or anxiety symptoms. The presence of children may also be protective against depressive and anxiety symptoms during lockdown. Having children may mitigate against loneliness that has been highlighted as a major issue during the isolation of lockdown.[26] A subset of the essential worker category–healthcare workers responsible for taking care of individuals with COVID-19—was associated with greater severity of problem drinking behaviours. Thus, the specific impact of lockdown on the necessity for essential workers and the impact of the burden of home schooling and childcare on parents appears to enhance drinking behaviours independently of an impact on psychiatric symptomatology.

As expected, having a personal relationship with someone who had become severely ill or died due to COVID-19 was associated with a greater increase in severity of problem drinking behaviours. Going outdoors more frequently for work, exercise, or essential duties during lockdown was similarly associated with greater severity of alcohol use, as well as depressive and anxiety symptoms. The reasons behind the need to go outdoors complicate the interpretation, as it might be confounded by being an essential worker, but would also allow for greater access to the purchase of alcohol. Living with others but having a poor quality of relationship was unexpectedly associated with lower drinking severity but with greater depressive and anxiety symptoms. Living alone was not associated with any changes in drinking behaviours but was associated with greater depressive symptomatology. These findings might support the role of drinking in the context of social interactions and further highlight the importance of socialisation during lockdown, the role of loneliness, and its impact on mental health.[26] Importantly, those residing in the UK–unlike those in the USA and Canada—displayed an increase in weekly alcohol units consumed during quarantine, consistent with the WHO Global Status Report on Alcohol and Health (2018), which shows that total alcohol consumed per capita is higher in the UK than in the USA or Canada.[27]

We further observed a relationship between the current severity of drinking behaviours and psychiatric symptoms such as depression, anxiety, and positive urgency. These relationships were driven particularly by the group that increased their drinking during quarantine. That both negative and positive emotionality factors are associated with increased drinking behaviours is in keeping with the multiple paths towards problematic alcohol use. The effects of depression and anxiety on alcohol consumption in both AUD and non-AUD drinkers are well documented,[28–31] and related to mechanistic theories of negative emotionality, which suggest that individuals may increase their alcohol consumption in stressful contexts to cope with aversive emotional states.[32] Positive emotional factors also appear to play a role in the association with positive urgency, a subtype of impulsivity characterised by the propensity to engage in disinhibited behaviours- including alcohol consumption- when experiencing an intensified hedonic or excited state.[31] Positive affect-based impulsivity may reflect a heightened reward sensitivity associated with problem drinking behaviours.[33]

## Limitations and future directions

This study is not without limitations. HabiT is a cross-sectional, retrospective survey and hence potentially limited by recall and misclassification biases as well as lack of longitudinal follow-up. Because retrospective reporting involves issues with memory, possible Dunning-Kruger effects, and selection bias, the reader should be cautious in drawing causal interpretations from the current data. Because the aim of the HabiT study was to investigate changes in amount and severity of drinking behaviour in a large, wider population, we issued the survey internationally and during a later period of enforced isolation. Thus, the possibility cannot be overlooked that subjects were within varying phases of lockdown characterised by differential restrictions during the time of testing, which may have influenced our current results. Future studies may consider data analysis by country, level of lockdown, or amount and severity of localised COVID-19 cases. Also, approximately half of the individuals who began the survey did not complete it. This may be due to the length of the survey (ie, 8–10 min). Prospective studies using an online survey design should further condense questionnaires and/or offer subjects monetary incentives obtained on survey completion in order to attenuate dropout and non-response bias. The current HabiT survey only assessed the *acute* effects of COVID-19 isolation measures on changes in drinking behaviours in comparison to the prequarantine period. Hence, follow-up studies should be employed during the immediate postquarantine period to investigate the possible protracted effects of COVID-19 isolation on drinking behaviours. Furthermore, whether the sampling adequately reflects the population distribution in the form of sampling bias may be an issue with online questionnaires and may under-represent those who do not have smartphones or access to the internet,[34] have limited facility with online questionnaires (eg, older individuals),[34] were otherwise engaged (eg, caring for an ill individual or children), or are more severely ill with substance use or other mental health disorders. Thus, our ability to

generalise our current findings to the wider population is limited. Other methods (eg, phone surveys) are recommended to reach populations under-represented by online surveys.[35] As few respondents reported a previous history of alcohol problems relative to the expected prevalence rates, the reporting is likely either a function of sampling bias, limited willingness to reveal such a history in an online survey, or marked changes in alcohol use particularly if relapse occurs. This limits our capacity to assess the change in drinking behaviours in those with a history of alcohol problems. Further studies focusing specifically on the newly abstinent or those with a history of alcohol problems are indicated.

## CONCLUSION

Although alcohol drinking behaviours appeared to decrease overall during lockdown, we emphasise that specific groups may be at higher risk for developing problematic alcohol use. In particular, factors associated with an increase in alcohol use include older individuals, essential workers, parents with children, those with a personal relationship with someone severely ill from COVID-19, and those with higher depression, anxiety, or positive urgency impulsivity. Furthermore, unlike residents from the USA and Canada, those in the UK increased their weekly alcohol intake during the quarantine period. We emphasise that those with a previous history of alcohol misuse or a family history of AUD were not the specific focus of this study, and may represent a high risk group that requires further investigation. Alcohol can be used in brief, moderate amounts in a healthy, non-pathological manner related to socialisation and stress relief. However, a subgroup of these individuals may still be at higher risk for longer term issues with alcohol misuse. The lockdown resulted in a unique set of stressors that in some cases may persist (eg, childcare, grieving, prolonged depression or anxiety related to the lockdown) and might again re-emerge with the imposition of localised lockdowns or further lockdowns in the context of a second or third wave. Further studies on the longitudinal impact and persistence of these behaviours are critical. Our findings highlight a need for identifying those at greater risk for alcohol misuse to aim for greater support services and proactively target mental health issues associated with problem drinking behaviours such as depression or anxiety.

**Contributors** SNS created the Habit Tracker (HabiT) survey, collaborated with VR in analysing the collected data, and drafted and edited the manuscript. VR coded and analysed the data. HB-J collaborated with VV in conceptualising the study. VV conceptualised the study, gave crucial guidance in creating the HabiT survey and edited the manuscript.

**Funding** This research was registered as a no-cost project, under grant number G107438. VV is supported by a MRC Senior Clinical Fellowship (MR/P008747/1).

**Competing interests** None declared.

**Patient consent for publication** Not required.

**Ethics approval** HabiT was approved by the Cambridge Psychology Research Ethics Committee.

**Provenance and peer review** Not commissioned; externally peer reviewed.

**Data availability statement** All participant data used in this research are deidentified. Participant data and MATLAB statistical code used for analysis are available on reasonable request from corresponding author, Samantha N Sallie, at habittstudy2020@gmail.com.

**ORCID iD**
Samantha N Sallie http://orcid.org/0000-0003-0161-3995

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
