## [Reviewer comments · BMJ Open]

ARTICLE DETAILS

TITLE (PROVISIONAL)	Assessing International Alcohol Consumption Patterns During Isolation from the COVID-19 Pandemic Using an Online Survey: Highlighting Negative Emotionality Mechanisms
AUTHORS	Sallie, Samantha; Ritou, Valentin; Bowden-Jones, Henrietta; Voon, Valerie

VERSION 1 – REVIEW

REVIEWER	Associate Professor Ben Edwards Australian National University, Australia
REVIEW RETURNED	08-Sep-2020

GENERAL COMMENTS	This paper examines the alcohol consumption during the COVID-19 pandemic via an online survey. As I see it there are a few major limitations with this paper centred around the survey methodology. My main concern is the highly selective sample. Specifically only 1,346 of 2,873 were used based on selection criteria. Beyond the selection criteria I would like to see a logistic regression model based on available data with dummy variables. Moreover, there was little information on the extent to which online survey was biased which is highly. Were there other auxiliary information, questions such as how did you find out about the survey? Similarly to what extent did COVID impact on abstention from alcohol use? It is unclear from the paper and an important finding on alcohol use would be the extent to which taking up alcohol a response to stress? Another important opportunity lost is that there is a lack of sub-sample analyses. As noted in the introduction males and females have very different alcohol consumption levels and the sample size affords sub-sample analyses. More importantly country specific analyses of residents from the US and UK should be conducted, or at least country specific dummies should be included in statistical analyses. A further missed opportunity is the timing of the lockdown in countries and the completion of the survey. This information should be incorporated into the statistical modelling, to understand the nature of the stress. While the limitations of a selective sample and retrospective reporting are acknowledged, the implications should be further discussed because in it's absence it is very difficult to interpret key findings.
--

REVIEWER	Prof Charles Parry South African Medical Research Council. South Africa
REVIEW RETURNED	10-Sep-2020

GENERAL COMMENTS	The paper addresses an important and timely question that of how social isolation measures in the midst of the Covid-19 pandemic may have affected drinking behaviour in the general adult population. The secondary aim of investigating if Covid-19 related stress factors influenced changes in drinking amount, drinking severity, depression, and anxiety during the quarantine is also highly pertinent. These aims/objectives were clearly motivated in the Introduction. The authors describe an appropriate research methodology which appears to have been well executed, and the resulting data appropriately analysed, presented and written up. The findings are of substantial interest to the emerging literature on drinking, mental health and life in the midst of pandemics with severe lockdown measures. I have a number of comments, some of which are of a more minor nature:  1. Only 46.8% of the participants provided usable data. This is a limitation. It would be useful to know more about the people who did not supply usable data so as to allow the reader to assess if they might reflect certain ages, genders, and persons from certain countries. Statistical tests could be provided to assess whether there are significant differences between the participants who provided usable and non-usable data on these variables. 2. While it is acknowledged that 789 or 58.6% of the participants with usable data came from the UK or USA, no details are given about the countries from which over 40% of the respondents came. More details should be provided and possibly even some analyses done by country even if only using data from the UK, USA and 1 or 2 other countries to see whether key findings are robust across countries as a kind of sensitivity analysis. This would deal with the concern that it is not appropriate to mix data from different countries with possibly very different lockdown circumstances, Covid-19 infection rates, and possibly even different drinking practices. Without details of the countries, the lockdown periods in each, and their individual drinking profile the reader does not know if it is appropriate to put all the data in the same pot for analyses. It would be helpful to include such information in an appendix. Country drinking data can be obtained from the WHO Global Status Report on Alcohol and Health (2018). 3. It was not entirely clear why the authors included a measure of impulsivity. This could be better motivated. 4. Introduction, page 5, line 9 – perhaps elaborate on amenities (i.e. food or medical care). 5. Introduction, page 5, line 42. Add a reference. 6. Methods, Covid-19 related stress scale. While there was a question on change in employment status due to Covid-19, there is no question specifically looking at substantial impact on financial status. Just wondering why not. 7. Results, demographic information (page 10, line 14): Possibly add “(63.8%)” after “859” so readers can see the % who are current drinkers. 8. Results, page 10, line 20. Add in numbers from other countries. 9. Results, page 10, lines 23-25 I think the £ sign is missing and a note that this is ANNUAL income.
--

	10. It might be useful to have a table giving the demographic details for drinkers, non-drinkers and the whole sample. If that is not possible, at least give details separately for drinkers so one can know more about this important sub-population. 11. Results, page 10, line 37 is it 5.62 +/-9.55 units PER WEEK? 12. Discussion, limitations section, page 16. The authors acknowledge that the participants were in different phases of lockdown with different restrictions. This would have substantially affected the findings and that is why I suggest in Bullet 2 that sub-analyses be done by countries with more data, e.g. UK and USA. See also bullet 11 below. 13. Discussion, page 17, line 12. It should be stated more clearly that the findings might be biased by the kind of people who agree to participate in online surveys. It is probably that people without smartphones and older people are less likely to do this. Refer to literature on this and add to limitations section. 14. Under future research section, suggest analyzing findings by country or level of lockdown/severity of Covid-19 cases. Possibly also suggest using other methods to reach populations not reached by online surveys such as doing phone surveys.
--	--

VERSION 1 – AUTHOR RESPONSE

Reviewer 1

As I see it there are a few major limitations with this paper centred around the survey methodology. My main concern is the highly selective sample. Specifically only 1,346 of 2,873 were used based on selection criteria. Beyond the selection criteria I would like to see a logistic regression model based on available data with dummy variables. Moreover, there was little information on the extent to which online survey was biased which is highly. Were there other auxiliary information, questions such as how did you find out about the survey?

We did not include any questions related to how individuals found out about the survey, although, in hindsight, this would have been preferable. Here, we assumed that the large majority of individuals either heard of the survey through one of the news outlets which advertised the survey or through family and friends. If we re-circulate this survey during a potential second lockdown or proximal post-lockdown period, we will include a question of this nature to track and mitigate possible sampling bias. Although we understand the potential for the logistic regression analysis model, we are using statistical tests to control for differences such as the country- and gender-specific variables. We hope this is satisfactory.

Similarly to what extent did COVID impact on abstention from alcohol use? It is unclear from the paper and an important finding on alcohol use would be the extent to which taking up alcohol a response to stress?

Thank you, this is a very important point. We have now included in the section entitled “Overall changes in drinking frequency and severity before and during quarantine”, line 9, the amount of subjects and percentage of the sample who reported consuming 1+ unit(s) of alcohol during a week in November and 0 units during the week of the quarantine period. We believe this to be a sufficient proxy of alcohol abstention during quarantine. We have reported this as follows: “Further, 172 (20%) subjects reported abstention from alcohol consumption during the quarantine period.”

Another important opportunity lost is that there is a lack of sub-sample analyses. As noted in the introduction males and females have very different alcohol consumption levels and the sample size

affords sub-sample analyses.

Yes, we believe you are absolutely correct about including sub-sample analyses by gender. We were limited by space but have now included, in the appendix, separate analyses for both males and females by employing the statistical procedure used on the overall sample in relation to change in weekly amount, change in severity, and overall current severity. We believe this will provide more clarity to our readers related to gender-specific differences in drinking patterns before and during COVID-19 quarantine.

More importantly country specific analyses of residents from the US and UK should be conducted, or at least country specific dummies should be included in statistical analyses. A further missed opportunity is the timing of the lockdown in countries and the completion of the survey. This information should be incorporated into the statistical modelling, to understand the nature of the stress.

This is a very important point, thank you for mentioning it. To your comment, we have included sub-sample country level analyses in the appendix of our manuscript. We have focused on the change in weekly drinking amount, change in drinking severity, and current overall severity of the UK, US, and Canada [as Canadian residents provided the third most data (n=64)]. We hope this addition is sufficient in allowing readers to interpret our findings derived from the overall sample.

While the limitations of a selective sample and retrospective reporting are acknowledged, the implications should be further discussed because in its absence it is very difficult to interpret key findings.

We agree that this is a research design limitation that should be more thoroughly discussed in the manuscript. With a selective sample, it is difficult to generalise findings to the wider population, and retrospective reporting involves issues with memory, possible Dunning-Kruger effects, and several other issues. We have added a more detailed description of the issues in lines 2-5 of the Limitations and future directions section of the manuscript, highlighted in blue.

Reviewer 2

1. Only 46.8% of the participants provided usable data. This is a limitation. It would be useful to know more about the people who did not supply usable data so as to allow the reader to assess if they might reflect certain ages, genders, and persons from certain countries. Statistical tests could be provided to assess whether there are significant differences between the participants who provided usable and non-usable data on these variables.

Yes, we agree this is a major limitation to the generalisability of the findings of this study. To your comment, we have provided in the appendix demographic (age and gender) analyses of those who dropped out of the study. The mean age of dropouts was significantly younger than those who completed the survey and more males dropped out than females or other genders. We believe this may reflect the larger demographic of the survey, as more males than females or others participated in (and completed) the survey.

2. While it is acknowledged that 789 or 58.6% of the participants with usable data came from the UK or USA, no details are given about the countries from which over 40% of the respondents came.

More details should be provided and possibly even some analyses done by country even if only using data from the UK, USA and 1 or 2 other countries to see whether key findings are robust across countries as a kind of sensitivity analysis. This would deal with the concern that it is not appropriate to mix data from different countries with possibly very different lockdown circumstances, Covid-19 infection rates, and possibly even different drinking practices. Without details of the countries, the lockdown periods in each, and their individual drinking profile the reader does not know if it is appropriate to put all the data in the same pot for analyses. It would be helpful to include such information in an appendix. Country drinking data can be obtained from the WHO Global Status Report on Alcohol and Health (2018).

Thank you for this insightful comment. We agree that it is important to further address the issue of international data analysis due to the late period of quarantine by which the data was collected. Due to this issue, we have conducted country level analyses for the top three participating countries (US, UK, and Canada), located in the appendix. We have found that, overall, change in and severity of drinking behaviours in the UK were higher than that observed in the US and Canada during the quarantine period. This is consistent with the WHO Global Status Report on Alcohol and Health (2018), now cited in the discussion, which shows that total alcohol per capita consumption (APC) is higher in the UK than in the US or Canada.

3. It was not entirely clear why the authors included a measure of impulsivity. This could be better motivated.

As impulsivity is widely accepted as a psychological feature predisposing individuals to problematic substance usage and addiction, we believed it was relevant to include a measure of it in our study. Further, as the impulsivity measure that we used also assesses mood-based impulsivity (or impulsive behaviours arising from intensified positive or negative emotional states), we believe this serves as a medium between other measures of impulsivity assessed in the questionnaire and the depression and anxiety symptomology centred on in our COVID-19 stress factor evaluation. We have added more details about our motivation in the introductory section of the manuscript, found in lines 28-29.

4. Introduction, page 5, line 9 – perhaps elaborate on amenities (i.e. food or medical care).

In the introduction, we have included some examples of basic amenities in lines 4-5.

5. Introduction, page 5, line 42. Add a reference.

We have now added a reference to this area of the paper.

6. Methods, Covid-19 related stress scale. While there was a question on change in employment status due to Covid-19, there is no question specifically looking at substantial impact on financial status. Just wondering why not.

This is a great comment. In retrospect, we should have included a question related specifically to COVID-19 isolation impact on financial status. If we are to recirculate the survey during a second wave lockdown or during a proximal post-isolation period, we will include a question of this nature.

7. Results, demographic information (page 10, line 14): Possibly add "(63.8%)" after "859" so readers can see the % who are current drinkers.

We have included the percentage of individuals in the sample who reported they drink alcohol in the results section line 4.

8. Results, page 10, line 20. Add in numbers from other countries.

We have included the numbers of subjects from the two countries which included the most respondents (after the UK and US), which were Canada (n=64) and Germany (n=63) in the demographics portion of the results section, highlighted in blue.

9. Results, page 10, lines 23-25 I think the £ sign is missing and a note that this is ANNUAL income.

As the survey was international, we asked individuals to report their amount earned yearly in their own currency, which may or may not be sterling pounds. However, we have more clearly specified that this is a measure of annual income in lines 8-9 of the demographics section, highlighted in blue.

10. It might be useful to have a table giving the demographic details for drinkers, non-drinkers and the whole sample. If that is not possible, at least give details separately for drinkers so one can know more about this important sub-population.

Thank you, this is a great suggestion in order for readers to more explicitly understand who composes our sample in relation to who consumes alcohol and who does not. We have now included a table in the appendix which provides the demographic information of those who reported drinking alcohol. This information includes age (mean, sd, range), gender, country total and amount of those residing in the UK and US, annual income, relationship status, and psychiatric condition (depression, anxiety, PTSD, comorbid depression and anxiety).

11. Results, page 10, line 37 is it 5.62 +/-9.55 units PER WEEK?

Yes, this is correct. We have now specified this change is “per week” in line 37 of the results section.

12. Discussion, limitations section, page 16. The authors acknowledge that the participants were in different phases of lockdown with different restrictions. This would have substantially affected the findings and that is why I suggest in Bullet 2 that sub-analyses be done by countries with more data, e.g. UK and USA. See also bullet 11 below.

Yes, this is a very important point. We have now included sub-sample analyses by country (US, UK, and Canada) to the appendix of the manuscript, as well as addressed in the Limitations and future directions section of discussion. More detailed information about the modification of this modification to the manuscript can be found under points 2 and 14.

13. Discussion, page 17, line 12. It should be stated more clearly that the findings might be biased by the kind of people who agree to participate in online surveys. It is probably that people without smartphones and older people are less likely to do this. Refer to literature on this and add to limitations section.

Thank you for the very helpful comment which helps to clarify this point to readers. We have further described types of individuals who are less likely to participate in online surveys (e.g., individuals without smartphones and older individuals). We have cited the relevant literature on this matter in lines 16-21 of the “Limitations and future directions” section, and hope this is a sufficient clarification.

14. Under future research section, suggest analyzing findings by country or level of lockdown/severity of Covid-19 cases. Possibly also suggest using other methods to reach populations not reached by online surveys such as doing phone surveys.

Thank you for these important comments in relation to recommendations for future research. Although, as in your comments 2 and 12, we have conducted country level sub-analyses of the three countries with the most subjects participating in the study (provided in the appendix), we have also included both points in the Limitations and future directions section (i.e., country level analyses: lines 7-8; phone surveys: lines 20-21), highlighted in blue.

VERSION 2 – REVIEW

REVIEWER	Ben Edwards Australian National University, Australia
REVIEW RETURNED	22-Oct-2020

GENERAL COMMENTS	Thank you for further sub-sample analyses and further details about the limitations of the study. One final suggestion is to provide the Oxford Stringency Index scores for each country as a guide to the level of lockdown in the main countries for sub-sample analyses.
---

REVIEWER	Charles Parry South African Medical Research Council, South Africa
REVIEW RETURNED	25-Oct-2020

GENERAL COMMENTS	The authors appear to have adequately addressed all but one of the 14 concerns/comments I made in my earlier review. Under demographic information data is presented on annual income, <19.9K (is this in UK pounds, converted for countries like Canada, USA and Germany?). What is the currency? Page 11, line 6, remove "in" before "0.89".
--

VERSION 2 – AUTHOR RESPONSE

Reviewers' Comments to Author:

Reviewer: 1

Thank you for further sub-sample analyses and further details about the limitations of the study. One final suggestion is to provide the Oxford Stringency Index scores for each country as a guide to the level of lockdown in the main countries for sub-sample analyses.

Thank you very much for your previous and current suggestions for our manuscript. We believe our manuscript has been greatly improved due to them. We have now included Oxford stringency index scores during time of data collection for the three countries which underwent subsample analyses in our study, as well as included total amount of confirmed COVID-19 cases and COVID-related deaths. This can be found in the supplemental materials (although it is referenced in the main body of the paper on page 10, lines 13-16).

Reviewer: 2

The authors appear to have adequately addressed all but one of the 14 concerns/comments I made in my earlier review. Under demographic information data is presented on annual income, <19.9K (is this in UK pounds, converted for countries like Canada, USA and Germany?). What is the currency? Page 11, line 6, remove "in" before "0.89".

Thank you for all of your previous and current comments regarding our manuscript; they were incredibly helpful. On page 11, line 6, we have removed the word "in" before the "0.89." Thank you for pointing out this grammatical error. Further, we have specified in the demographics section, line 11, that annual income was presented to participants in the survey as raw currency (i.e., whichever currency was relevant to their country of residence) but converted to sterling (i.e., UK) pounds during analysis, highlighted in blue.